# Diabetes regulates fructose absorption through thioredoxin-interacting protein

**James R Dotimas**[1,2†], **Austin W Lee**[1,2†], **Angela B Schmider**[3], **Shannon H Carroll**[1,2], **Anu Shah**[1,2], **Julide Bilen**[1,2], **Kayla R Elliott**[1,2], **Ronald B Myers**[1,2], **Roy J Soberman**[3,4], **Jun Yoshioka**[1,2], **Richard T Lee**[1,2*]

[1]Department of Stem Cell and Regenerative Biology, Harvard University, Harvard Stem Cell Institute, Cambridge, United States; [2]Department of Medicine, Brigham and Women's Hospital and Harvard Medical School, Cambridge, United States; [3]Nephrology Division, Department of Medicine, Massachusetts General Hospital, Charlestown, United States; [4]Molecular Imaging Core, Massachusetts General Hospital, Charlestown, United States

**Abstract** Metabolic studies suggest that the absorptive capacity of the small intestine for fructose is limited, though the molecular mechanisms controlling this process remain unknown. Here we demonstrate that thioredoxin-interacting protein (Txnip), which regulates glucose homeostasis in mammals, binds to fructose transporters and promotes fructose absorption by the small intestine. Deletion of *Txnip* in mice reduced fructose transport into the peripheral bloodstream and liver, as well as the severity of adverse metabolic outcomes resulting from long-term fructose consumption. We also demonstrate that fructose consumption induces expression of Txnip in the small intestine. Diabetic mice had increased expression of Txnip in the small intestine as well as enhanced fructose uptake and transport into the hepatic portal circulation. The deletion of *Txnip* in mice abolished the diabetes-induced increase in fructose absorption. Our results indicate that Txnip is a critical regulator of fructose metabolism and suggest that a diabetic state can promote fructose uptake.

**\*For correspondence:**
Richard_Lee@harvard.edu

[†]These authors contributed equally to this work

**Competing interests:** The authors declare that no competing interests exist.

## Introduction

The dietary consumption of fructose has drastically increased in modernized societies (*Cox, 2002*), and growing evidence implicates fructose consumption in contributing to the worldwide increase in metabolic diseases, such as nonalcoholic fatty liver disease, obesity, and type 2 diabetes mellitus (*Johnson et al., 2007*; *Ouyang et al., 2008*). The uptake of fructose by the small intestine controls its availability to organs that are able to metabolize fructose, including liver (*Asano et al., 1992*) and kidney (*Sugawara-Yokoo et al., 1999*). Following ingestion, fructose is absorbed by enterocytes in the small intestine, first entering from the intestinal lumen through glucose transporter 5 (GLUT5) on the apical membrane (*Burant et al., 1992*) and then exiting the enterocyte into the bloodstream through glucose transporter 2 (GLUT2) on the basolateral membrane (*Gould et al., 1991*), although GLUT2 can translocate to the apical membrane in response to increased fructose levels in the lumen (*Gouyon et al., 2003*; *Kellett and Brot-Laroche, 2005*).

Following transport through the small intestine, fructose travels through the hepatic portal vein to the liver, where it is phosphorylated to fructose 1-phosphate by hepatic fructokinase, bypassing the energy-sensitive enzymatic activity of phosphofructokinase, which generates fructose 1,6-biphosphate in the glucose metabolic pathway (*Mayes, 1993*). Downstream metabolic intermediates of fructose, including acetyl-CoA, can be directed toward *de novo* lipogenesis, promoting nonalcoholic fatty liver disease (*Abdelmalek et al., 2010*). The build-up of hepatic triglycerides also promotes

**eLife digest** Fructose is a type of sugar that is found naturally in fruits, and it is closely related to glucose. The amount of fructose in our diet has increased dramatically in the last few decades. Growing evidence suggests that excessive amounts of fructose contribute to several metabolic diseases, including fatty liver disease and diabetes. Fructose is absorbed in the small intestine via transport proteins called GLUT2 and GLUT5 and then travels to the liver where it can stimulate the cells to make fats. However, it is not clear how fructose uptake is regulated in the small intestine.

Glucose is taken into cells by a transport protein that is closely related to GLUT2 and GLUT5. Another protein called thioredoxin-interacting protein (Txnip) interacts with the glucose transporter and regulates glucose uptake. Here, Dotimas et al. investigated whether Txnip also regulates the activities of GLUT2 and GLUT5 to control how cells absorb fructose. Initial experiments in cells showed that Txnip binds to both GLUT2 and GLUT5 and increases the amount of fructose taken up by both mouse and human cells.

Cells from mutant mice that do not produce Txnip absorbed less fructose than normal cells did. Furthermore, the mutant mice had lower levels of fructose in the blood and less severe metabolic disease after consuming fructose regularly for six months. Mice with diabetes absorbed more fructose through the small intestine than normal mice, and the loss of Txnip from these mice abolished this effect.

Together the findings of Dotimas et al. suggest that Txnip plays an important role in regulating fructose absorption and indicate that, at least in some circumstances, diabetes may lead to more fructose being absorbed in the small intestine. The next steps following on from this work are to understand the molecular details of how Txnip regulates fructose uptake and to determine if other forms of diabetes also show increased fructose uptake.

insulin resistance and obesity by impairing insulin signaling and increasing global lipid circulation (*Wei and Pagliassotti, 2004*). Compared to blood glucose levels (in the range of 5 mM in humans), blood fructose levels are maintained at low levels (*Douard and Ferraris, 2008*) (0.008 to 0.5 mM in humans) through the efficient clearance by the liver and, to a lesser extent, by the kidneys (*Mayes, 1993*). Thus, the pathogenesis of fructose-associated metabolic disease is dependent on the function of GLUT2 and GLUT5 as the primary transporters of fructose of enterocytes on the small intestine, particularly the duodenum and jejunum, which facilitate most carbohydrate absorption.

Txnip, or thioredoxin-interacting protein, is an arrestin-like protein that can bind to thioredoxin protein and that regulates metabolism in mammals (*Shalev, 2014*; *Patwari and Lee, 2012*). We have previously reported that Txnip overexpression represses cellular glucose uptake while eliminating Txnip expression increases glucose uptake in peripheral tissues in both insulin-dependent and insulin-independent manners (*Parikh et al., 2007*). Expression of Txnip is highly correlated with extracellular glucose concentrations which upregulate the activity of the transcription complexes chREBP/Mlx and MondoA/Mlx that bind to the carbohydrate response element (ChoRE) on the *Txnip* promoter to induce *Txnip* mRNA expression (*Cha-Molstad et al., 2009*; *Stoltzman et al., 2008*). Upregulated Txnip inhibits glucose uptake (*Patwari and Lee, 2012*) by interacting with and altering the expression of glucose transporter 1 (GLUT1) (*Wu et al., 2013*). Because Txnip regulates glucose transport and fructose metabolism may be a significant factor in important metabolic diseases, we studied the effect of Txnip on fructose absorption and on fructose-associated metabolic disease.

## Results

### Txnip interacts with fructose transporters GLUT2 and GLUT5 and promotes fructose uptake

Given that Txnip is a regulator of glucose homeostasis (*Parikh et al., 2007*; *Patwari et al., 2009*), we sought to explore its potential to regulate fructose metabolism, specifically through uptake by the small intestine. In order to determine potential molecular interactions between Txnip and GLUT2

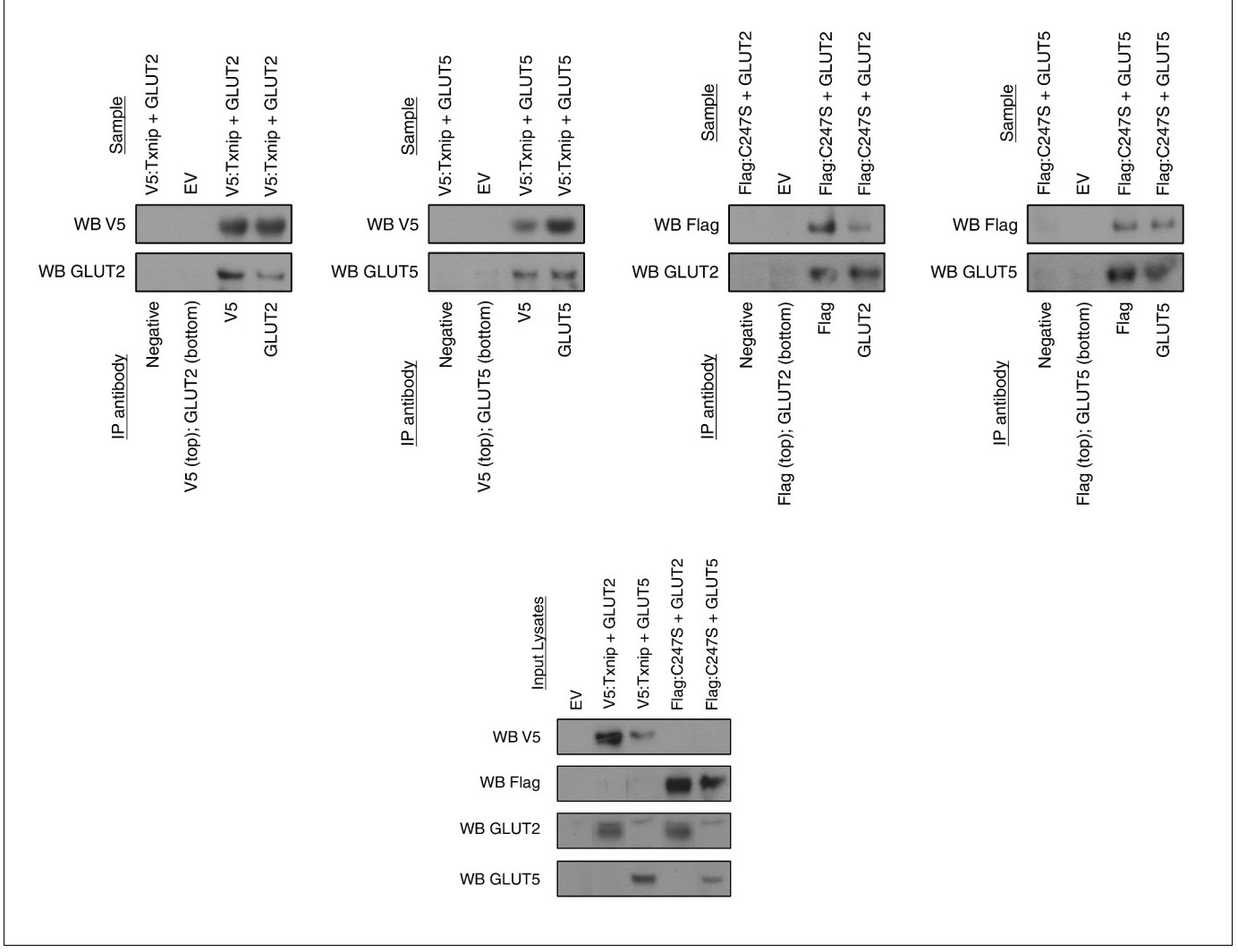

**Figure 1.** Txnip binds to GLUT2 and GLUT5 independently of its thioredoxin-interacting cysteine. hTXNIP-V5 or its C247S-flag mutated form were co-expressed with either hGLUT5 or hGLUT2 in HEK293 cells. Cellular lysates were captured using antibodies that were then bound to Protein A/G agarose beads. The input and captured complexes were then immunoblotted for the proteins of interest. The results indicate that Txnip binds to both GLUT2 and GLUT5 and that the *Txnip* mutant C247S, which abolishes the molecular interaction between Txnip and thioredoxin, can still bind GLUT5 and GLUT2.

The following source data is available for figure 1:

**Source data 1.** Images represent the cropped Western Blot presented in the manuscript on the left-hand side accompanied by the developed film from which it was cropped on the right-hand side.

or GLUT5, we performed a pulldown assay by co-immunoprecipitating TXNIP with either GLUT2 or GLUT5. We observed an interaction between TXNIP and both GLUT2 and GLUT5 even in the absence of its C247 thioredoxin-interacting site (*Kellett and Brot-Laroche, 2005*) (*Figure 1*), indicating that binding to thioredoxin is not necessary for TXNIP to interact with GLUT2 or GLUT5.

Having established a molecular interaction between Txnip and both transporters that mediate fructose uptake, we then used Caco-2 cells transiently transfected with human TXNIP, GLUT2, and GLUT5 overexpressing plasmid to determine if Txnip expression affects the cellular uptake of 14C-radiolabeled D-fructose. The Caco-2 cell line is a human epithelial colorectal adenocarcinoma cell line that when plated as a confluent monolayer resembles the enterocyte lining of the small intestine, both morphologically and functionally (*Hidalgo et al., 1989*; *Engle et al., 1998*). Measuring the

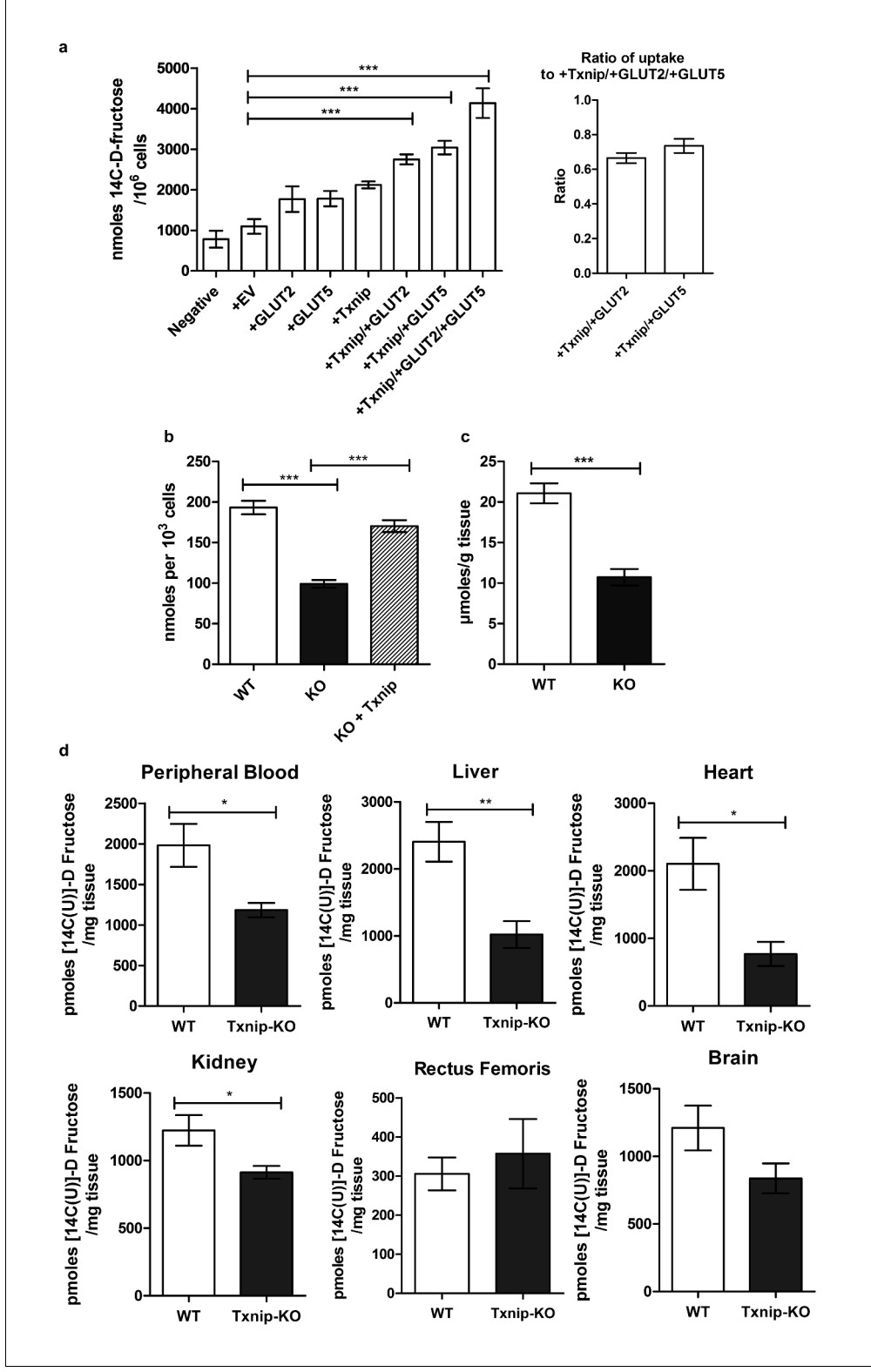

**Figure 2.** Txnip promotes cellular fructose uptake and fructose absorption by the small intestine. (a) Empty vector (EV) or *TXNIP*, with *GLUT2* and/or *GLUT5*, was transiently transfected in Caco-2 cells. Simultaneous TXNIP-overexpression with either GLUT2 or GLUT5 overexpression induced an increase in cellular fructose uptake relative to EV-transfected cells while the greatest increase in uptake was observed in cells overexpressing TXNIP with both GLUT2 and GLUT5. There was no significant difference in uptake in cells overexpressing TXNIP with GLUT2 in *Figure 2 continued on next page*

*Figure 2 continued*

comparison to cells overexpressing TXNIP with GLUT5 (n = 6). (**b**) High passage mouse embryonic fibroblasts (MEFs) from *Txnip*-KO animals had decreased cellular fructose uptake compared with cells from wild type mice, which was restored by transfecting *TXNIP* in the same cell line (n = 6). (**c**) Jejunum from *Txnip*-knockout mice is able to transport less fructose from intestinal lumen when exposed to [14C(U)]-D fructose compared to small intestine from wild type mice. (n = 4). (**d**) Following administration of a [14C(U)]-D fructose bolus via oral gavage, *Txnip*-knockout mice exhibit reduced 14C signal in the blood compared to wild type mice. Other organs, including the liver, kidney, and heart, showed the same trend of reduced 14C signal in *Txnip*-knockout mice relative to wild-type mice, though the signal in kidney and heart was lower than that of blood. There was no observed difference in the skeletal muscle or brain within the parameters of the experiment (n = 4). *p<0.05, **p<0.01, ***p<0.001. Data represent mean ± SEM.

The following source data and figure supplements are available for figure 2:

**Source data 1.** Statistical analysis of *Figure 2*.
**Source data 2.** Statistical analysis for *Figure 2—figure supplement 1*.
**Figure supplement 1.** Wild type and *Txnip*-null mouse had no difference in organ 14C D-fructose uptake following intravenous injection.
**Figure supplement 2.** Determination of oral gavage single time point.

retention of 14C-radiolabeled D-fructose in confluent Caco-2 cells revealed that simultaneous over-expression of TXNIP with GLUT2 or GLUT5 increased cellular 14C-radiolabeled D-fructose uptake by 161% and 162% (p<0.01, p<0.001; *n* = 6) respectively relative to the empty vector (EV) transfection control (*Figure 2a*). Interestingly, overexpression of either fructose transporter in the absence of TXNIP overexpression was insufficient to produce a marked increase in cellular fructose uptake, suggesting that Txnip expression is required to facilitate the function of GLUT2 and GLUT5 under some circumstances. Overexpression of TXNIP with both fructose transporters produced the greatest increase in cellular fructose uptake. After normalizing the uptake in cells overexpressing TXNIP/ GLUT2 or TXNIP/GLUT5 to that of cells overexpressing all three proteins, we found that the normalized ratios of uptake were not different in cells overexpressing TXNIP/GLUT2 and TXNIP/GLUT5, suggesting that the effects of GLUT2 and GLUT5 may be similar and synergistic in Txnip-mediated fructose uptake. A reciprocal loss-of-function experiment using high-passage mouse embryonic fibroblasts (MEFs) isolated from WT or *Txnip*-null (*Txnip*-KO) mice revealed that *Txnip*-KO MEFs retained 51.3% of the fructose compared to WT MEFs (p<0.001; *n* = 6); this reduction in cellular uptake was rescued by transient overexpression of human TXNIP, showing that the deletion of *Txnip* reduces cellular fructose uptake (*Figure 2b*). An ex vivo analysis revealed a reduced uptake of 14C-radiolabeled D-fructose of 50.9% (p<0.001; *n* = 6) in jejunum isolated from *Txnip*-KO mice compared to WT mice, indicating that Txnip expression promotes fructose uptake by the small intestine, particularly by the jejunum, which, in conjunction with the duodenum, is the primary site of fructose absorption (*Figure 2c*).

To evaluate if the reduction in small intestinal fructose uptake caused by the deletion of *Txnip* regulates the levels of fructose in the peripheral bloodstream and other organs *in vivo*, we performed an oral gavage of 14C-radiolabeled D-fructose in WT and *Txnip*-KO mice. We administered 14C-radiolabeled fructose in a 30% solution of fructose and mannitol (to correct fructose uptake into tissue for the adherent extracellular fluid phase) and allowed for 20 min of digestion before measuring the retention of 14C-radiolabeled fructose in various tissues of interest. We used 20 min as the standard time point for this experiment, as it was the time point at which the peripheral blood in wild type mice contained the greatest 14C signal (*Figure 2—figure supplement 2*), which was also confirmed in our later experiments (Figure 5). The liver from *Txnip*-KO mice demonstrated a reduction of 14C-radiolabeled D-fructose to 42.4% compared to liver from WT mice (p<0.05; *n* = 4), suggesting that the transport of fructose from the intestinal lumen to the hepatic portal blood system was reduced in *Txnip*-KO mice (*Figure 2d*); this finding was confirmed in additional data shown below in measurements of hepatic portal vein blood (Figure 5). The reduced levels of 14C-

radiolabeled fructose received and processed by the liver subsequently manifested in peripheral blood in *Txnip*-KO mice, which had a fructose signal that was 54.9% compared to that of WT (p<0.01; *n* = 4). (*Figure 2d*). Heart and kidney tissue from *Txnip*-KO mice also exhibited a decrease in 14C D-fructose signal, but this signal was lower than that of the peripheral blood, indicating that the observed difference may be due to decreased circulating levels of fructose and not by uptake by these organs. Interestingly, the rectus femoris muscle from both groups demonstrated no differences within the parameters of the experiment and had a fructose signal lower than levels found in circulating blood despite previous studies reporting fructose uptake and Txnip expression in skeletal muscle (*Parikh et al., 2007*; *Zierath et al., 1995*) (*Figure 2d*). Brain tissue collected from the cerebrum also did not exhibit a difference in 14C D-fructose signal. These results indicate that the deletion of *Txnip* reduces fructose absorption and subsequent availability to other organs in vivo.

To test if the absence of Txnip expression also affects fructose uptake by other organs independent of differences in intestinal uptake, we measured 14C D-fructose signal in several tissues collected 30 min following the intravenous tail injection of 14C-radiolabeled D-fructose, effectively bypassing the small intestine and hepatic portal vein circulation. After normalizing the 14C D-fructose signal of the other organs to that found in the blood, we observed no differences in any of the organs collected, including liver and kidney, both of which have the ability to absorb and process fructose (*Figure 2—figure supplement 1*). These results indicate that the effect of Txnip on fructose absorption may be specific to the small intestine. As such, the small intestine may be the primary regulator of fructose access to the body, or other compensatory mechanisms may exist to regulate fructose concentrations in these organs.

## Txnip is necessary for fructose-associated metabolic disorders

While the physiological consequences of dietary fructose consumption remain controversial, there is increasing evidence suggesting an association between fructose consumption and metabolic diseases (*Johnson et al., 2007*; *Ouyang et al., 2008*; *Stanhope, 2016*). To determine if the negative effects of chronic fructose consumption are diminished in *Txnip*-KO mice, which have a marked decrease in fructose absorption, we supplemented a moderate fat diet (containing 0.16% fructose) of WT and *Txnip*-KO mice with 30% w/v fructose in their drinking water; this diet is similar to the Western-style diet, which is enriched with saturated fats and sugar (*Stevenson et al., 2016*). We maintained mice on the fructose-supplemented diet (FSD) for 25 weeks before measuring differences in physiological parameters in comparison to mice on the moderate fat regular diet (RD). Because male mice on a moderate fat diet develop adverse metabolic outcomes even without supplementary fructose (*Morselli et al., 2014*; *Hwang et al., 2010*), we analyzed the effects of the fructose-supplemented diet on female mice. WT mice on the FSD had a higher body weight of 7.73 ± 2.61 g relative to WT mice on the RD (mean ± SEM; *n* = 5, p<0.05), while *Txnip*-KO mice on both diets maintained similar weights to the WT mice on the RD (*Figure 3a*). *Txnip*-KO mice have reduced fasting blood glucose levels and are able to effectively maintain decreased blood glucose levels following insulin injection and reduce blood glucose levels to baseline levels following glucose injection (*Chutkow et al., 2008*). The prolonged FSD caused significant systemic intolerance to intraperitoneal injections of glucose (2 g/kg) as well as reduced response to insulin (0.25 milliunits/g) in WT animals (*n* = 5), whereas *Txnip*-KO mice were resistant to these metabolic consequences caused by the FSD (*Figure 3b,c*). Sirius Red staining of liver sections used to measure hepatic fibrosis revealed that WT mice on the FSD had a nearly five fold increased hepatic collagen content relative to the WT mice on the RD (*n* = 5, p<0.01), while *Txnip*-KO mice had comparable levels to WT mice on the RD regardless of the diet they were fed (*Figure 3d*). Liver sections graded using a previously described method for hepatocellular steatosis (*Burgess et al., 2011*) revealed that WT mice on the FSD scored significantly worse compared to mice on the RD (*n* = 5, p<0.001), indicating more liver fat content and steatosis; the severity of steatosis was significantly reduced in *Txnip*-KO mice on the FSD ([p<0.001] [*Figure 3e*]). Together, these data demonstrate that the *Txnip*-KO mice on the FSD had reduced severity of adverse metabolic outcomes associated with a high fructose diet relative to their WT counterparts. This phenotype may result from decreased absorption of fructose by the small intestine but could be attributed to the lack of Txnip in the various other tissues.

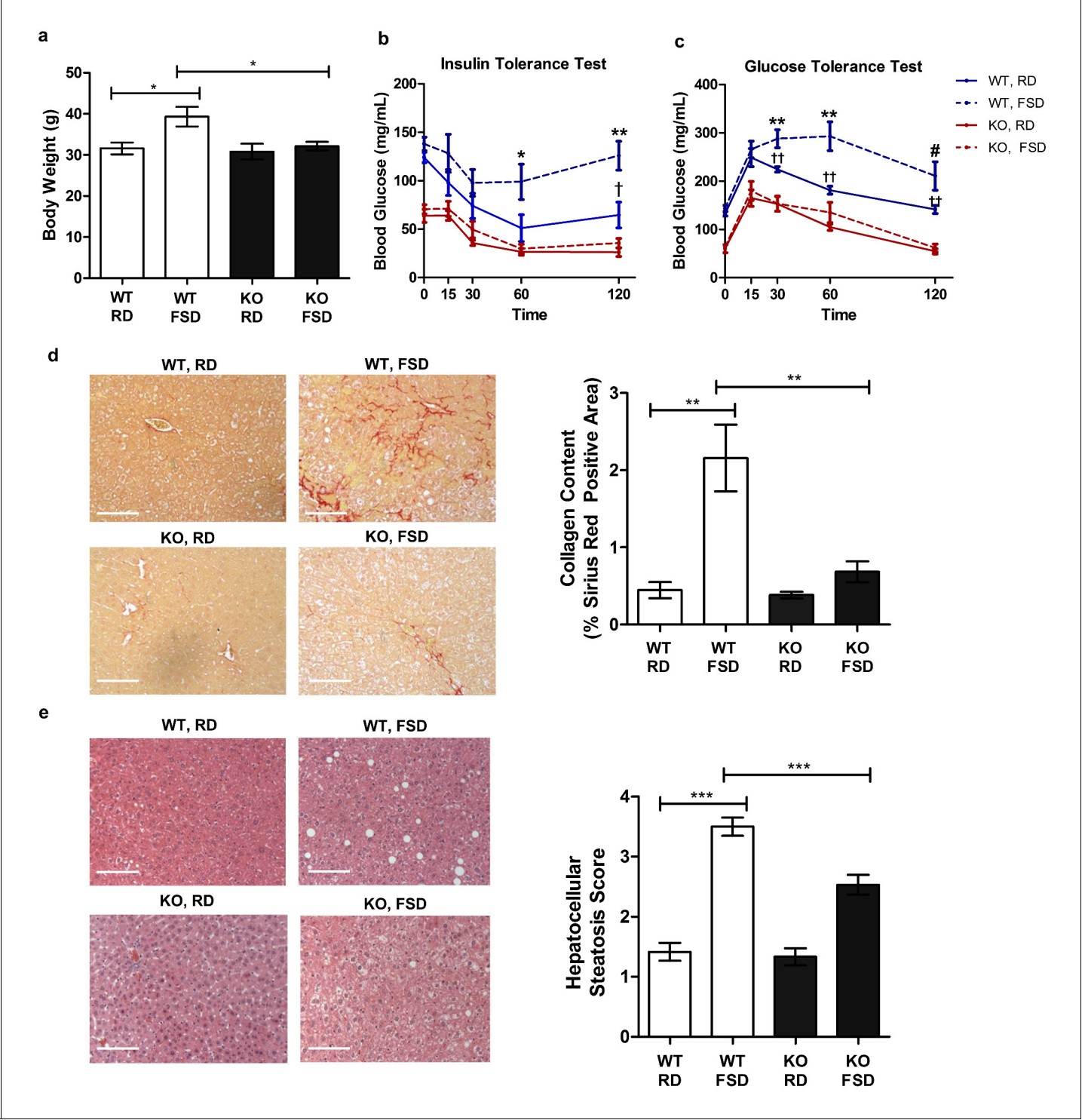

**Figure 3.** *Txnip*-knockout mice exhibit less severe metabolic outcomes associated with chronic fructose consumption. Mice were fed a moderate fat diet supplemented with 30% (wt/vol) fructose water for 25 weeks. (a) In comparison with the control diet, fructose induced significant increases in body weight in wild type (WT) but not in *Txnip*-KO mice. (b and c) Intraperitoneal glucose and insulin tests were performed. WT animals developed significant glucose and insulin intolerance following prolonged high fructose diet. *Txnip*-KO mice were resistant to these effects. (d and e) Histological analysis of the liver sections. (d) *Txnip*-KO liver had observably less fibrosis after fructose consumption compared with WT. (e) Fructose diet induced significant steatosis in WT mice that was reduced in *Txnip*-KO mice. (n = 5 per group). *p<0.05. For Glucose Tolerance Test and Insulin Tolerance Test: *p<0.05 and **p<0.01 WT, RD vs WT, FSD; †p<0.05 and ††p<0.01 WT, RD vs. KO, RD. #p<0.05 WT, RD vs. KO, FSD and p<0.01 vs WT, RD. Scale bar; 50 µm. Data represent mean ± SEM.

*Figure 3 continued on next page*

*Figure 3 continued*

The following source data is available for figure 3:

**Source data 1.** Statistical analysis for *Figure 3*.

## Fructose induces Txnip expression and interaction with fructose transporters in the small intestine

Previous studies demonstrated that Txnip expression not only regulates glucose homeostasis but also is responsive to glucose levels (*Shalev et al., 2002*). To determine if Txnip is similarly responsive to chronic fructose consumption, we investigated the effects of chronic fructose consumption on the expression levels of Txnip and its potential interactions with GLUT2 or GLUT5 in the jejunum, which, along with the duodenum, is responsible for fructose absorption by the small intestine (*Holloway and Parsons, 1984*). Quantitative RT-PCR analysis revealed that mRNA expression of *GLUT2* and *GLUT5* increased significantly in the jejunum of mice on the FSD relative to those on the RD, regardless of the genotypes of the mice (*Figure 4a*). In addition, WT mice on the FSD had higher mRNA expression of *Txnip* in the jejunum compared to the RD, indicating that Txnip expression is responsive to fructose consumption in a manner similar to the changes observed in GLUT2 and GLUT5 expression (*Figure 4b*). Confocal analysis of jejunum sections probed for Txnip and GLUT2 or GLUT5 revealed that Txnip co-localized with both GLUT2 and GLUT5 in WT mice on the RD (*Figure 4c,d*). The co-localizing signal increased in WT mice on the FSD, with the Txnip and GLUT2 co-localizing signal increasing by 12.9% ± 4.4% (mean ± SEM; $n$ = 5, p<0.02) of the total GLUT2 signal and the Txnip and GLUT5 co-localizing signal increasing by 6.0% ± 2.3% (mean ± SEM; $n$ = 5, p=0.05) within the total GLUT5 signal (*Figure 4b*).

We then analyzed the interaction between Txnip and GLUT2 or GLUT5 interacting molecules using Fluorescent Lifetime Imaging Microscopy (FLIM). Because two separate groups of mice were studied (normal diet and fructose diet) to determine the effect of fructose on the interaction between Txnip and GLUT2 and GLUT5, we analyzed FLIM values between Txnip and GLUT2 and GLUT5 within each group relative to donor only controls (Txnip only). We then compared these relative changes across groups. We measured FLIM values at co-localizing regions between Txnip and GLUT2 or GLUT5, and we measured an equal number of regions across all samples. As such, an increase in protein mass resulting from the observed mRNA expression is unlikely to significantly impact the results. Rather, we expect observed differences to result from changes in the nature of the interaction between Txnip and both fructose transporters. The values for both $\tau_1$ and $a_1$(%) were calculated for Txnip within one group. The averages of these values were used for comparison across groups. After determining the donor fluorescence lifetime (Txnip) in the absence of the acceptor fluorophore (*Table 1*, Donor Only), FRET between donor and acceptor was defined by the lifetime of interacting molecules ($\tau_1$), with $a_1$ (%) representing the fraction of interacting molecules. $\tau 1$ is an indirect measurement of distance between two molecules because the lifetime of the donor fluorophore diminishes with increasing proximity to the acceptor fluorophore, which can quench donor emissions. We found that Txnip interacted with both GLUT2 and GLUT5 in the small intestine of animals on the RD as indicated by a decrease in $\tau_1$ (*Table 1*). Under the FSD, we observed a decrease in $\tau_1$ for both GLUT2 and GLUT5 relative to samples form the RD, indicating increased proximity between Txnip and the two GLUTs (*Table 1*). This result indicates that the Txnip interaction with the fructose transporters responds to fructose loads. It also suggests that both fructose transporters are involved in Txnip-mediated fructose transport, consistent with our fructose uptake experiments (*Figure 2a*). Though $a_{1\%}$ values increased modestly for GLUT2 under the FSD relative to the RD, the contribution of this increase to fructose uptake is not clear (*Table 1*). Collectively, these data suggest that Txnip expression and interaction with GLUT2 and GLUT5 in the jejunum are responsive to chronic fructose consumption, and these changes may contribute to fructose absorption into the body.

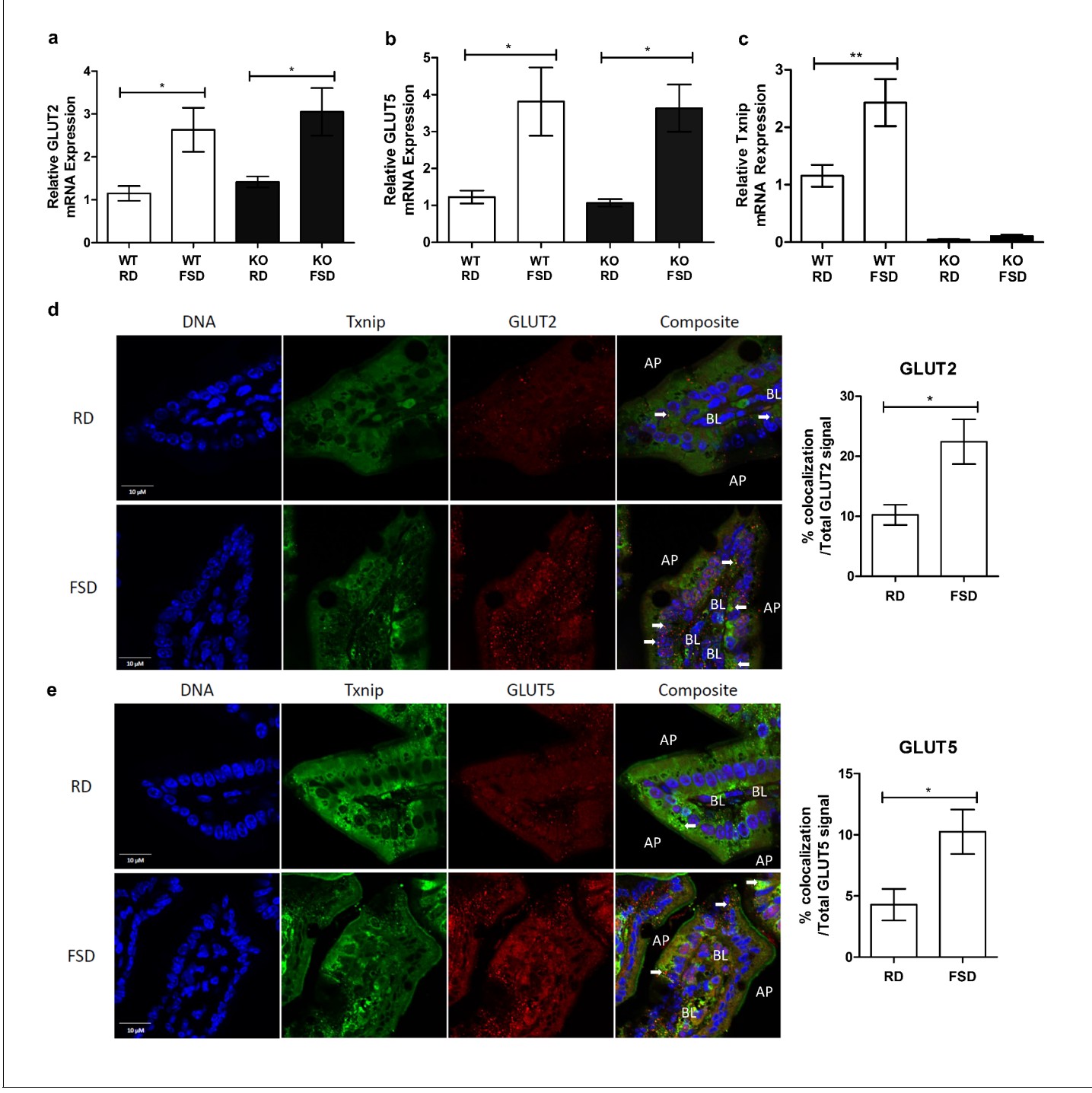

**Figure 4.** Txnip expression is induced by fructose consumption and interacts with GLUT2 and GLUT5. (a–c) Quantitative RT-PCR analysis of jejunum isolated from mice used in the fructose supplemented diet experiment. (a and b) Mice on the FSD had significantly higher mRNA expression of GLUT2 and GLUT5 in the jejunum ($n$ = 6 for WT, RD and WT, FDS groups; $n$ = 5 for KO, RD and KO, FSD groups). (c) Mice on the FSD had higher relative mRNA expression of *Txnip* in the small intestine ($n$ = 6 for WT, RD and WT, FDS groups; $n$ = 5 for KO, RD and KO, FSD groups). (d) Confocal analysis of small intestine sections stained for Txnip and GLUT2 revealed co-localization of the two proteins, with more co-localizing signal in FSD mice ($n$ = 4 for each group). (e) Confocal analysis of small intestine sections stained for Txnip and GLUT5 revealed co-localization of the two proteins, with more co-localizing signal in FSD mice ($n$ = 4 for each group). *$p < 0.05$, **$p < 0.01$. AP = apical membrane, BL = basolateral membrane, white arrows point to co-localizing signal, scale bar represents 10 µM. Data represent mean ± SEM.

The following source data is available for figure 4:

*Figure 4 continued on next page*

*Figure 4 continued*

**Source data 1.** Statistical analysis for *Figure 4*.

## Insulin-deficient diabetes induced by streptozotocin increases Txnip expression and fructose absorption

Txnip expression is elevated in skeletal muscle of individuals with impaired glucose tolerance or type 2 diabetes, and Txnip expression is inversely correlated with insulin-stimulated glucose uptake in human insulin/glucose clamp studies (*Muoio, 2007*). To determine if diabetes has a similar effect on Txnip expression in the small intestine, we used the streptozotocin (STZ) model of type 1 diabetes in mice. All mice receiving STZ injections developed hyperglycemia, allowing us to compare the effect of this diabetic phenotype to euglycemic sodium citrate buffer (vehicle)-injected non-diabetic controls. Although *Txnip*-KO mice are hypoglycemic compared to their wild type counterparts at baseline (*Chutkow et al., 2008*), STZ-injected *Txnip*-KO mice achieved a higher average blood glucose level relative to non-diabetic *Txnip*-KO mice (*Figure 5—figure supplement 1*). Quantitative RT-PCR revealed that WT STZ-injected mice had higher levels of *Txnip* mRNA expression in the jejunum compared to wild type non-diabetic mice, indicating that the diabetic state promotes Txnip expression in the small intestine (*Figure 5a*).

We then investigated whether STZ-injected mice had reduced rates of absorption and reduced total absorption by measuring the retention of 14C-radiolabeled D-fructose in hepatic portal vein

**Table 1.** Txnip/GLUT interactions determined by FLIM.

**Regular diet (RD)**

|  | Donor only | GLUT2 | GLUT5 |
|---|---|---|---|
| $a_1$ (%) Experiment 1 | 100 | 35 ± 2 | 32 ± 9 |
| Experiment 2 | 100 | 36 ± 4 | 36 ± 1 |
| Experiment 3 | 100 | 36 ± 2 | 35 ± 3 |
| Average $a_1$ (%) | N/A | 35 | 34 ± 1 |
| $\tau_1$ (ps) Experiment 1 | 2701 ± 33 | 2169 ± 43 | 2102 ± 48 |
| Experiment 2 | 2196 ± 15 | 1615 ± 60 | 1657 ± 26 |
| Experiment 3 | 2139 ± 36 | 1587 ± 30 | 1695 ± 31 |
| % of $\tau_1$ (ps) vs donor only | N/A | 76 ± 2 | 77 ± 1 |
| **Fructose-supplemented diet (FSD)** | | | |
|  | Donor only | GLUT2 | GLUT5 |
| $a_1$ (%) Experiment 1 | 100 | 49 ± 4 | 31 ± 5 |
| Experiment 2 | 100 | 45 ± 2 | 36 ± 3 |
| Experiment 3 | 100 | 55 ± 2 | 36 ± 3 |
| Average $a_1$ (%) | N/A | 50 ± 3[†] | 34 ± 2 |
| $\tau_1$ (ps) Experiment 1 | 2482 ± 35 | 1621 ± 51 | 1263 ± 99 |
| Experiment 2 | 2495 ± 62 | 1701 ± 38 | 1572 ± 53 |
| Experiment 3 | 2298 ± 72 | 1575 ± 10 | 1602 ± 34 |
| % of $\tau_1$ (ps) vs donor only | N/A | 67 ± 1* | 60 ± 6* |

Values represent mean percent change ± SEM.

*$p < 0.05$ vs. regular diet.

[†]$p < 0.01$ vs. regular diet.

**Source data 1.** Statistical analysis for *Table 1*. This table represents the statistical analysis conducted on the raw data collected for *Table 1* using GraphPad Prism 5.

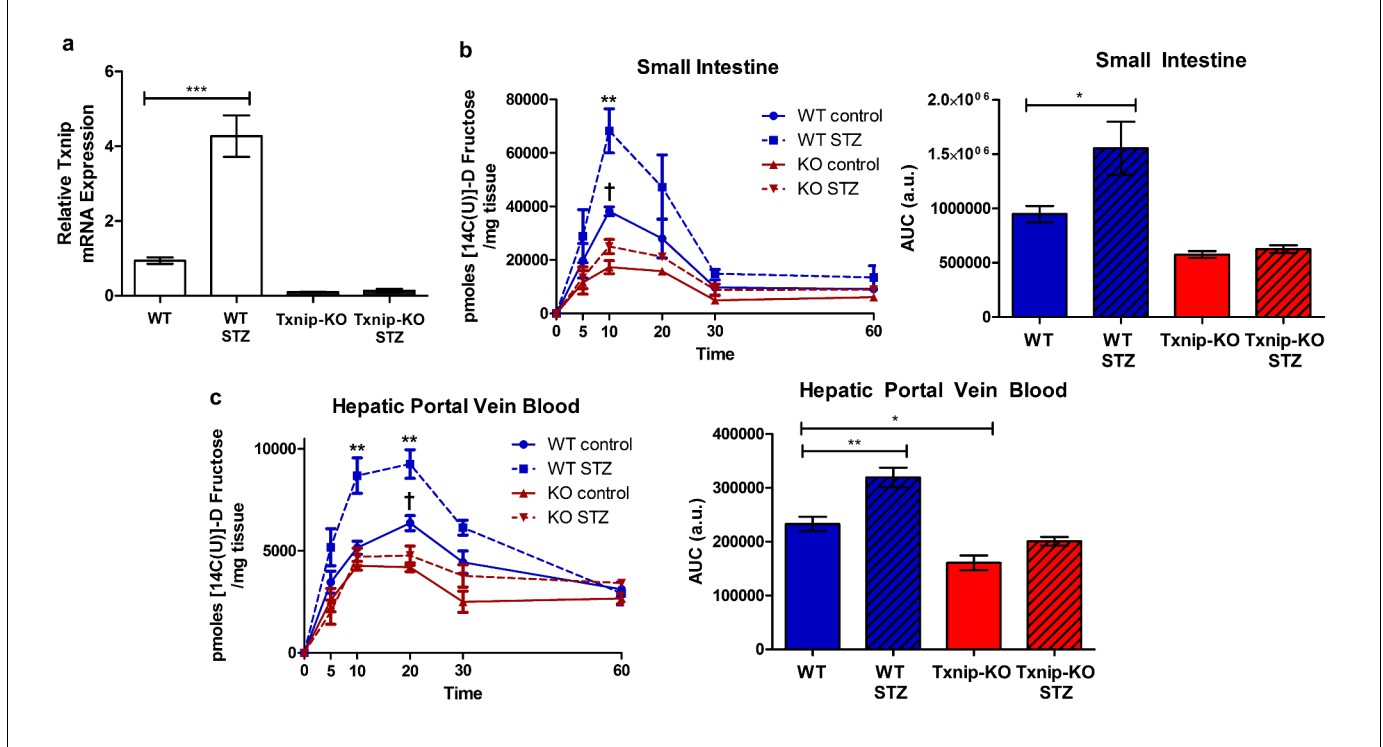

**Figure 5.** Hyperglycemia promotes Txnip expression and fructose absorption by the small intestine. (a) STZ-injected wild type mice have higher *Txnip* expression in the small intestine relative to control mice. (*n* = 4 for all groups). (b and c) 14C-radiolabeled D-fructose levels were measured in WT and *Txnip*-KO STZ-injected and buffer controls over time. Following an oral gavage, tissues were collected at 5, 10, 20, 30, 60 min post-gavage for analysis (*n* = 4 for all groups at each time point). (b) There was a significant increase in 14C fructose uptake by the whole small intestine in WT STZ-injected mice compared to the non-diabetic controls at 10 min post-gavage as well as a reduction in uptake in *Txnip*-KO non-diabetic mice compared to WT non-diabetic mice. There was no difference at any time point between *Txnip*-KO STZ-injected animals and their non-diabetic counterparts. Comparing overall levels of 14C fructose uptake, there was an increase in 14C fructose uptake in WT STZ-injected mice compared to the non-diabetic controls, while this increase was abolished in the *Txnip*-KO mice. (c) There was a significant increase in 14C fructose signal in the hepatic portal vein blood in WT STZ-injected mice relative to the non-diabetic controls at both 10 and 20 min post-gavage. There was also a decrease in 14C fructose signal in *Txnip*-KO non-diabetic mice compared to WT non-diabetic mice. When comparing overall levels of 14C fructose signal, there was a significant increase in WT STZ-injected mice compared to the non-diabetic controls; this difference was not observed between the *Txnip*-KO groups. ***p<0.001. For fructose absorption data: **p<0.01 WT control vs. WT STZ, †p<0.05 WT control vs. KO control. Data represent mean ± SEM.

The following source data and figure supplements are available for figure 5:

**Source data 1.** Statistical analysis for *Figure 5*.
**Source data 2.** Statistical analysis for *Figure 5—figure supplement 1*.
**Source data 3.** Statistical analysis for *Figure 5—figure supplement 2*.
**Figure supplement 1.** STZ-injection induces hyperglycemia in mice.
**Figure supplement 2.** 13C-labeled D-fructose measurements in wild-type non-diabetic and STZ-injected mice.

blood and whole small intestine at different time points following an oral gavage. Relative to non-diabetic WT mice, STZ-injected WT mice demonstrated an increase in 14C signal in the whole small intestine at 10 min after the administration of the 14C-D-fructose bolus, suggesting a greater rate of absorption (*Figure 5b*). There was no observable difference in 14C signal in whole small intestine collected from non-diabetic *Txnip*-KO mice relative to STZ-injected *Txnip*-KO mice, revealing that Txnip expression is necessary for the STZ-mediated increase in fructose absorption by the small intestine. When comparing overall levels of 14C signal over time, there was a significant difference

in total fructose absorption between the STZ-injected WT and non-diabetic WT mice, while this STZ-mediated effect was abolished in the *Txnip*-KO mice (*Figure 5c*). The increased levels of 14C-radio-labeled D-fructose measured in the small intestine subsequently manifested in increased observed amounts of 14C signal in hepatic portal vein blood collected from STZ-injected WT mice relative to non-diabetic controls at both 10 min and 20 min; this difference was absent when comparing STZ-injected *Txnip*-KO mice to non-diabetic *Txnip*-KO mice. Furthermore, there was an observable difference in overall 14C signal in the hepatic portal vein blood collected from STZ-injected WT mice compared to non-diabetic WT mice, while there was no difference between the *Txnip*-KO groups (*Figure 5c*).

In order to test that the observable differences were attributable to D-fructose and not a downstream metabolite of the administered 14C-fructose, we studied 13C-labeled D-fructose levels in the small intestine using LC-MS analysis. We collected jejunum from animals 10 min post-administration of the oral gavage and isolated the polar metabolites from the tissue for analysis. There was a marked increase in 13C-labeled D-fructose measured in the jejunum collected from STZ-injected mice relative to non-diabetic controls (*Figure 5—figure supplement 2*).

Collectively, these data demonstrated that STZ-injected mice have overall higher jejunum Txnip expression and fructose absorption, an effect that is diminished in the absence of Txnip expression. Thus, the STZ-induced diabetic state can increase fructose absorption, and Txnip is essential for this effect.

## Discussion

This study establishes that fructose absorption in mammals is regulated by Txnip, which interacts with both GLUT2 and GLUT5 on the cellular membrane of enterocytes in the small intestine. While it has long been known that glucose promotes fructose uptake by the small intestine, the exact mechanism for this effect remains unknown (*Riby et al., 1993*). It was once believed that glucose promotes fructose uptake through a disaccharide transport system (*Riby et al., 1993*). Our data in combination with other studies (*Parikh et al., 2007*; *Patwari et al., 2006*) suggests that Txnip links glucose homeostasis with fructose transport, as diabetes induces Txnip expression, which promotes fructose absorption. Because excess absorption of fructose contributes to liver fat accumulation and hypertension, our experiments suggest that the diabetic state may contribute to these components of metabolic disease at least in part through Txnip and increased fructose transport. The ability of Txnip to regulate fructose metabolism indicates that glucose homeostasis and fructose metabolism are intertwined, possibly in a manner that potentiates metabolic diseases in settings of high fructose consumption, which is often accompanied by similar loads of glucose as sucrose and others sweeteners.

The molecular mechanism through which Txnip regulates fructose absorption remains to be defined, and it is plausible that Txnip is necessary for the increase in fructose absorption in diabetic mice but is not directly causal. We demonstrated that Txnip interacts with both fructose transporters and is upregulated in response to fructose consumption. We also observed that the effect of Txnip on fructose transport is specific to the small intestine, as organs collected from intravenously injected *Txnip*-null animals had similar amounts of fructose to wild-type counterparts. However, defining the detailed molecular mechanism through which Txnip regulates fructose transporters is important. Elevated levels of fructose and glucose in the small intestine can cause GLUT2 to transiently translocate to the apical brush border membrane of enterocytes in order to further facilitate fructose transport, accounting for up to 60% of fructose absorption in animals consuming high amounts of sugar (*Gouyon et al., 2003*). The translocation of GLUT2 is affected by activation of protein kinase C and MAP kinase (*Helliwell et al., 2000*; *Kellett, 2001*), both of which also have the ability to activate Txnip expression and activity (*Li et al., 2014*, *2009*). Past studies observed that activation of the PKC signaling pathway by phorbol 12-myristate (PMA) and the influx of Ca2+ through L-type channels induces cytoskeletal re-arrangement, leading to the insertion of GLUT2 protein at the brush border membrane (*Kellett and Helliwell, 2000*; *Morgan et al., 2007*). In addition, Txnip expression is attenuated by the calcium channel blocker verapamil (*Xu et al., 2012*), suggesting that normally due to the influx of calcium, Txnip could be facilitating the movement of GLUT2 to the apical membrane as a scaffold protein that directly binds to the glucose transporters.

Another possibility is that the phosphorylation of Txnip may change its ability to bind to the fructose transporters and cause subsequent changes in fructose absorption. A previous study found that AMPK activation had a positive effect on GLUT1-facilitated glucose transport, which the researchers attributed to the change in the binding affinity of Txnip for GLUT1 (*Wu et al., 2013*). Regulation of GLUT2 and GLUT5-mediated fructose absorption could occur through a similar mechanism in which the binding of Txnip is necessary for this process.

These processes may be specific to gastrointestinal fructose absorption, as we noted that a global loss of Txnip affected fructose uptake in a variety of tissues from fed animals but not in those from intravenously injected animals. Studying the Txnip-mediated fructose absorption mechanism, thus, will require a more precise inquiry into how enterocyte Txnip expression affects fructose uptake by the digestive tract. Other possibilities exist for why this discrepancy occurs because the intravenous injection not only bypasses the small intestine but also the first-pass metabolism and uptake by the liver.

Regulation of fructose absorption may be closely linked to other metabolic processes, including glucose metabolism and insulin signaling. While elevated glucose levels promote Txnip expression, elevated insulin levels decrease Txnip expression (*Parikh et al., 2007*). Because chronic fructose intake is associated with impaired insulin signaling, it could facilitate the disruption of glucose homeostasis and the transition of prediabetes to type 2 diabetes through Txnip. In addition, fructose metabolism can lead to an increase in reactive oxygen species (ROS) and subsequent redox stress in cells (*Morgan et al., 2007*). Because Txnip is responsive to the redox environment of the cell, fructose may indirectly regulate Txnip expression through this mechanism. As a result, fructose could affect Txnip in a manner that disrupts glucose metabolism and further increase fructose absorption by promoting Txnip expression, similar to how glucose can regulate Txnip to change fructose metabolism.

As noted by our long-term fructose feeding experiment, fructose ingestion may facilitate the onset of metabolic disease, including insulin resistance and hepatic steatosis. The deletion of Txnip alleviated mice of these effects to an extent, suggesting that Txnip and other factors involved in the Txnip-mediated fructose absorption pathway could be potential therapeutic targets. However, much more understanding of this molecular pathway is necessary to determine if and when regulation of the pathway may be beneficial. We observed that the STZ-induced diabetic state causes a marked increase in fructose uptake by the small intestine and transport into hepatic portal vein circulation. STZ-diabetic mice showed an increase in absorption of fructose, which could contribute to metabolic dysfunction, as was the case in our long-term fructose supplemented diet experiment in which the mice developed pathologies compared to mice with relatively lower fructose consumption. STZ induces hypoinsulinemia and subsequent hyperglycemia mediated through pancreatic beta cell death, so further studies will be necessary to determine if all causes of hyperglycemia can regulate fructose absorption by the small intestine. Future studies in mice with cell-specific deletion of *Txnip* may reveal more detail on how Txnip regulates fructose transport. Controversy remains on the effect of the diabetic state on fructose absorption (*Stanhope, 2016*). One study in Japan observed an increase in serum fructose concentrations in patients with type 2 diabetes mellitus relative to healthy patients who had no diabetic symptoms (*Kawasaki et al., 2002*); these patients were all admitted to the same hospital and had their diets monitored and were fasting when blood serum was collected. However, another study in Finnish patients did not observe a difference in fructose serum levels when comparing type 2 diabetic patients to patients with no diabetic symptoms (*Pitkänen, 1996*). The patients in that study were not fasted prior to blood collection, and many of the diabetic patients had many diabetes-related complications affecting their health. Currently, it is difficult to determine based on available human data if the diabetic state and which specific physiological pathology of diabetes may affect fructose absorption in humans. Our data suggest that glucose homeostasis and fructose absorption interact. Given the potential importance of fructose metabolism in modern societies, more investigation into this phenomenon is warranted.

## Materials and methods

### Animal care and usage

All experiments were conducted in accordance with the Guide for the Use and Care of Laboratory Animals and approved by the Harvard Medical School Standing Committee on Animals. *Txnip* knockout mouse from a C57B1/6 background strain were developed in our laboratory (RRID:IMSR_ JAX:018313) (*Morselli et al., 2014*). Genotyping was performed by PCR on tail DNA using the following primers: OL4F2, 5'- CTT CAC CCC CCT AGA GTG AT –3'; P3F1, 5'-TTT CGT TTG GGT TTT CAA GC –3'; and P3R2, 5'-CCC AGA GCA CTT TCT TGG AC–3'. The number of mice required for the study was determined by using the ClinCalc Sample Size Calculator: a change of 33% for all phenotypes was assumed, and an alpha of 0.05 and statistical power of 90% were also used as parameters to estimate the appropriate sample size.

### Pulldown assay

Pulldown assays were performed as previously described (*Abdelmalek et al., 2010*). Briefly, indicated plasmids were transfected into HEK293T cells using Transit-293 transfection reagent (Mirus). Cells were lysed in 0.5% Triton X-100, 0.1% sodium deoxycholate,150 mM NaCl, 50 mM Tris, 1 mM phenylmethanesulfonyl fluoride, and protease and phosphatase inhibitors, pH 7.8. Immunoprecipitation of Txnip protein complexes was performed using the appropriate antibodies before binding to Protein A/G agarose beads (Santa Cruz Biotechnology; Dallas, Texas). Input lysates and pulldown eluates were analyzed by SDS–PAGE and immunoblots. The following protein-specific antibodies were used: anti-V5 (RRID:AB_307024), anti-flag (RRID:AB_298215), anti-GLUT2 (RRID:AB_641068), and anti-GLUT5 (RRID:AB_2189499). The following HRP-conjugated antibodies were used: Goat Anti-Mouse IgG (H L)-HRP Conjugate (RRID:AB_11125936) and Rabbit Anti-Mouse IgG (Light Chain Specific) (D3V2A) mAb HRP Conjugate (RRID:AB_1549610).

### In vitro fructose uptake assay

Caco-2 cells or isolated MEFs were transfected with the indicated plasmids using Purefection Transfection Reagent (System Biosciences) and incubated for 24 hr. Fructose was added to normal medium conditions at a concentration of 4.5 g/L with the addition of 5 µCuries/mL of D-[14C]fructose (MP Biomedicals). Cells were incubated with fructose for 1 hr before lysis in 0.5% Triton X-100, 150 mM NaCl, 50 mM Tris, 1 mM phenylmethanesulfonyl fluoride, and protease and phosphatase inhibitors, pH 7.8. Cell lysates were measured with a gamma counter (Beckman Coulter LS 6500).

### Intestinal ex vivo uptake measurements

Fructose uptake rates in the small intestine were determined following the technique of by Karasov and Diamond (*Hwang et al., 2010*). Briefly, a 1-cm segment of jejunum was everted and mounted on a grooved steel rod (3- mm diameter) and preincubated at 37°C for 5 min in Ringer solution bubbled with 95% O2-5% CO2. The sleeves were then incubated at 37°C in an oxygenated solution containing D-[14C]fructose for 2 min. The solutions were stirred at 1200 rpm during the incubation procedure to minimize unstirred layers. To reduce the radioactive label in the adherent fluid, there was a 20-s rinse in 30 ml of ice-old Ringer solution after incubation. The tissues were dissolved in a tissue solubilizer (Solvable, Packard Instruments). The dissolved tissue was mixed with scintillation cocktail (Ecolume, ICN), and radioactivity was measured with a liquid scintillation counter (Beckman LS 7800, Beckman, Fullerton, CA). The uptake rates of 14C D-fructose were determined at 50 mM and expressed as micromoles per milligram net weight of small intestine.

### In vivo biodistribution study with radiolabeled fructose

Intestinal fructose transport and its biodistribution were analyzed in vivo. After intragastric administration of D-Fructose, [U-14C] (0.2 µCurie in 200 µl 30% fructose/mannitol) with a ball tip needle or intravenous administration through tail-vein injection, mice were euthanized. Organs of interest (blood, small and large intestine, liver, kidney, and femur muscle) were harvested. Portal vein blood was extracted from the vein using heparinized tubing and needle. Approximately 50–150 mg of each tissue of interest was dissolved in Soluene-350 and added to the appropriate amount of Ultima Gold Scintillation Fluid. Radioactivity in tissues was then measured in the gamma-counter and results

were analyzed as percentage of injected dose per gram of tissues as previously described (*Chutkow et al., 2008*).

## Fructose diet

Txnip deficient mice (6-week-old, female) and wild type mice were divided into two groups. All animals were fed the PicoLab Mouse Diet 20. Mice consumed approximately 30–50 g of feed per day per mouse, amounting corresponding to around 20–33 kcals per day per mouse, with no significant differences in feed intake among the different groups. One group had free access to 30% (wt/vol) fructose water with otherwise regular diet, and the other group had free access to plain water with regular diet for 25 weeks. Mice with access to fructose on average consumed an additional 1 kcal per day that accounted for roughly 3–5% of their total caloric intake, which is similar to the average caloric contribution of fructose in humans (*Burgess et al., 2011*). Body weight, food intake, and water intake were measured. At the end of protocol, animals were sacrificed to harvest tissues. Measurements were normalized to organ weight and expressed as picomoles per mg of tissue.

## STZ injections

The DiaComp protocol for low dose streptozotocin induction in mice was used. Mice were fasted 4–6 hr prior to injections. STZ dissolved in 0.1 M sodium citrate (pH 4.5) was administered to each mouse at a concentration of 50 mg/kg mouse body weight. Injections were repeated daily for 5 days. Hyperglycemia was confirmed 2 weeks post-injections using a Bayer Contour Blood Glucose Meter. STZ injections were repeated after 1 month in mice that remained euglycemic.

## Glucose/insulin tolerance test

Glucose (2 g/kg) and insulin (2 milliunits/g) tolerance tests were performed intraperitoneally following 4 hr fasting period to measure plasma glucose levels at 30, 60, 120, and 180 min after injection. Whole blood glucose levels were assayed from tail clipping venous blood samples using an Ascensia Elite XL glucometer (Bayer Co).

## Histology

Liver samples were collected from mice and fixed in 10% formalin. Samples were then dehydrated using a gradient of 70–100% ethanol and 100% xylene. Tissues were embedded, sectioned, and stained with H&E and Sirius Red by the Harvard Stem Cell Institute Histology Core.

## Quantitative RT-PCR analysis

Relative gene expression was measured using qPCR. RNA was collected from jejunum samples from mice using the PureLink RNA Micro Kit (Thermo Fisher Scientific). RNA samples were then converted to cDNA using a High-Capacity cDNA Reverse Transcription Kit (Thermo Fisher Scientific). cDNA Samples were then measured using a Bio-Rad CFX384 Real-Time PCR Detection System using iTaq Universal SYBR Green Supermix (Bio-Rad) and the indicated primers (*Supplementary file 1*). All biological replicates were measured using technical triplicates.

## Confocal analysis

Jejunum from WT mice from the fructose diet experiment were harvested and embedded in paraffin. Sections were prepared by the Harvard SCRB Histology Facility. After rehydration and antigen recovery of the sections, slides were stained with GLUT2 (RRID:AB_641066) and GLUT5 (RRID:AB_2189502) with anti-rabbit secondary antibody conjugated to Alexa Fluor 594 (RRID:AB_2534079) and Txnip (RRID:AB_11033580) with anti-rabbit antibody conjugated to Alexa Fluor 488 (RRID:AB_143165). Slides were analyzed using an Olympus Fluoview 1500 at the Brigham and Women's Regenerative Medicine Center. Co-localization was analyzed using the co-localization tool in the FSV-10W software using the default threshold (intensity of 2024) for all samples and channels. Brightness and contrast were applied to equally to all images. All images were normalized to equal gamma levels.

## FRET-FLIM analysis

To analyze the interaction of Txnip with GLUT5 by Fluorescence Lifetime Microscopy (FLIM), Nikon Ti-E inverted microscope was used. Becker and Hickl SPCM software with DCC was used to acquire FLIM data, and SPCImage 3.0 (Becker and Hickl) software was used for FLIM analysis. Paraffin-embedded sections of small intestine from wild type mice on either the regular or fructose-supplemented diet were used. After rehydration and antigen recovery of the sections, Txnip was detected with secondary antibody conjugated to Alexa Fluor 488 (donor fluorophore) and GLUT2 or GLUT5 with antibody conjugated to Alexa Fluor 594 (acceptor fluorophore). Becker and Hickl SPCM software with DCC was used to acquire FLIM data, and SPCImage 3.0 (Becker and Hickl) software was used for FLIM analysis. The donor fluorophore was stimulated with a 488 nm laser, and the lifetime ($\tau_1$) was determined in the presence and absence of antibody to the acceptor fluorophore. The parameter $a_1$ (%) was automatically determined for a measurement of the percent of Txnip molecules interacting with GLUT2 or GLUT5 in the region being analyzed.

## LC-MS analysis

Tissues were prepared as indicated by the Metabolite Profiling Core at the Whitehead Institute. Approximately 10–30 mg of each tissue was collected from mice and immediately frozen using liquid nitrogen. Samples were homogenized in LC/MS grade methanol at $-20°C$ before adding LC/MS grade water and HPLC-grade chloroform. Samples were then spun down to separate the polar and organic metabolites. Samples were dried using a vacuum concentrator before submission to the Metabolite Profiling Core for analysis.

## Statistical analysis

All sample sizes reported refer to biological replicates. Statistical comparison between two groups was performed by unpaired Student's $t$ test. Statistical comparison among three or more groups was performed by one-way ANOVA with the Bonferroni post-hoc test of significance. Two-way ANOVA with the Bonferroni post-hoc test of significance was used to compare the means of groups in the fructose diet experiment and STZ-induction experiments. Statistical analysis was carried out with the Graphpad Prism 5 software (RRID:SCR_002798), and statistical significance was assigned to differences with a p value of <0.05.

## Acknowledgements

This study was performed with financial support from the National Institute of Health (RTL DK107396, AG040019 and HL117986, RJS 1R01AR065538, 1R01CA193520, R01DK062472, S10RR027931, and ABS K01DK089145).

## Additional information

### Funding

| Funder | Grant reference number | Author |
| --- | --- | --- |
| National Institutes of Health | AG040019 | Richard T Lee |
| National Institutes of Health | HL117986 | Richard T Lee |
| National Institutes of Health | 1R01AR065538 | Roy J Soberman |
| National Institutes of Health | S10RR027931 | Roy J Soberman |
| National Institutes of Health | K01DK089145 | Angela B Schmider |
| National Institutes of Health | DK107396 | Richard T Lee |
| National Institutes of Health | 1R01CA193520 | Roy J Soberman |
| National Institutes of Health | R01DK062472 | Roy J Soberman |

The funders had no role in study design, data collection and interpretation, or the decision to submit the work for publication.

## Author contributions

JRD, AWL, ABS, JY, Conception and design, Acquisition of data, Analysis and interpretation of data, Drafting or revising the article, Contributed unpublished essential data or reagents; SHC, AS, JB, Conception and design, Acquisition of data, Analysis and interpretation of data, Drafting or revising the article; KRE, RBM, Acquisition of data; RJS, RTL, Conception and design, Analysis and interpretation of data, Drafting or revising the article, Contributed unpublished essential data or reagents

## Author ORCIDs

Richard T Lee, http://orcid.org/0000-0003-4687-1381

## Ethics

Animal experimentation: This study was performed in strict accordance with the recommendations in the Guide for the Care and Use of Laboratory Animals of the National Institutes of Health. All of the animals were handled according to approved institutional animal care and use committee (IACUC) protocols (#03-723). The protocol was approved by the Committee on the Ethics of Animal Experiments of Brigham and Women's Hospital (Animal Welfare Assurance Number: A3431-01).

# Additional files

**Supplementary files**

• Supplementary file 1. Primers used for quantitative RT-PCR analysis.

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
