## [Decision Letter]

Thank you for submitting your article "Diabetes regulates fructose absorption through thioredoxin-interacting protein" for consideration by *eLife*. Your article has been reviewed by three peer reviewers, and the evaluation has been overseen by Mark McCarthy as the Senior Editor and Reviewing Editor. The following individuals involved in review of your submission have agreed to reveal their identity: John Horowitz (Reviewer #1); Anath Shalev (Reviewer #2).

The reviewers have discussed the reviews with one another and the Reviewing Editor has drafted this decision to help you prepare a revised submission.

Summary:

Dotimas et al. test the hypothesis that Txnip affects fructose metabolism. In reductionist systems, Txnip interacts with Glut5 and Glut2, and its expression promotes fructose transport. Cells from Txnip null mice have decreased fructose uptake, which is rescued by human Txnip. Gavage demonstrates decreased content of label presented as fructose in several tissues of Txnip KO mice, but these differences were not seen with tail vein administration of labeled fructose. Txnip deficient female mice eating a fat diet supplemented with fructose were protected from weight gain and abnormal glucose metabolism and protected from morphologic evidence of metabolic liver disease. The dietary intervention is associated with an interaction between Txnip and Glut5/Glu2 in the jejunum. Strep-induced diabetes increases Txnip expression in WT mice and appeared to promote fructose uptake.

This is an interesting study, which convincingly establishes that TXNIP is upregulated by fructose in the diet, and in turn promotes intestinal fructose absorption, potentially creating a positive feedback loop in those with diabetes in particular. There is also evidence that fructose absorption contributes to the development of diabetic liver disease. These are novel and provocative findings that speak to the possible contribution of dietary fructose with respect to the development of T2D. We felt that the current work falls short in providing a complete elaboration of the mechanisms underlying Txnip-induced fructose absorption, but this limitation is discussed and addressing it experimentally would likely lie beyond the scope of this paper.

Essential revisions:

1) The causative role of the Txnip-GLUT2/5 interaction in the processes described remains less clear. Theoretically, this could represent an unrelated scenario by which the identified protein-protein interaction does not contribute to the observed changes in fructose absorption. Can the authors address this issue? (Reviewer 2)

2) Can the authors comment on how the observed intestinal specificity might be conferred and what is the relative contribution of GLUT2 vs. GLUT5? (Reviewer 2)

3) Please address the following issues: Figure 2 presents data showing decreased label (14C) in several tissues of Txnip KO mice after gavage of labeled fructose at a single time point. How was this time point determined? Given the rapid metabolic rate in mice, substantial compartmental flux likely takes place over this time period. The fructose would be metabolized and signals might not necessarily reflect fructose transport. The authors address this problem (and its relevance to the strep-induced state) with the data of Figure 5—figure supplement 2. But these data do not quantitatively account for fructose absorption, this is done at 10 minutes after oral gavage, and the results are confusing. The text in the subsection “Insulin-deficient diabetes induced by streptozotocin increases Txnip expression and fructose absorption” indicates that these findings represent stable isotope label from jejunum, but the legend of Figure 5—figure supplement 2 indicates that the signal is from jejunum and hepatic portal vein blood. Why pool these compartments?

4) Please address the following: Confocal images in Figure 4 purporting to show greater interaction between Txnip and the glucose transporters with the FSD could be more compelling. This interaction is confirmed using FLIM, but these changes are not robust, even if they are statistically significant. Is it possible that this interaction is due to a trivial reason, the presence of increased glucose transporter protein mass? Figure 4 presents data showing that FSD increases message for Glut2 and Glut5. Are Glut2 and Glut5 protein abundances increased? If so, more interaction would be expected. What is missing from this line of research is insight into how Txnip might promote fructose transport. At least for other transporters, protein mass is not the major predictor of facilitated movement of sugars across membranes. Instead, the fusion of transporter vesicles with the plasma membrane is critical. Does Txnip mediate this fusion event?

---

## [Author Response]

*This is an interesting study, which convincingly establishes that TXNIP is upregulated by fructose in the diet, and in turn promotes intestinal fructose absorption, potentially creating a positive feedback loop in those with diabetes in particular. There is also evidence that fructose absorption contributes to the development of diabetic liver disease. These are novel and provocative findings that speak to the possible contribution of dietary fructose with respect to the development of T2D. We felt that the current work falls short in providing a complete elaboration of the mechanisms underlying Txnip-induced fructose absorption, but this limitation is discussed and addressing it experimentally would likely lie beyond the scope of this paper.*

We thank the reviewers for the thoughtful review of our manuscript. We also agree that the paper lacks a thorough inquiry into the molecular mechanism underlying the control of fructose absorption by Txnip, and here we discuss our current plans to investigate this phenomenon. In the following, we have addressed the reviewer specific points.

Essential revisions:

1) The causative role of the Txnip-GLUT2/5 interaction in the processes described remains less clear. Theoretically, this could represent an unrelated scenario by which the identified protein-protein interaction does not contribute to the observed changes in fructose absorption. Can the authors address this issue? (Reviewer 2)

We acknowledge that although we observed Txnip is necessary to promote fructose absorption, the molecular details of this process require a greater inquiry. While we have no experimental data to support a direct causal role of the Txnip-GLUT interaction in the observed changes in fructose absorption, we believe there are potentially plausible mechanisms.

In previous findings from our lab, we found through mass spectrometry of Txnip protein-protein interactions that Txnip binds to Rab11a (Lee et al., EMBO Mol Med 2014). Recent findings from Patel et al.suggest that Rab11a endosomal trafficking of GLUT5 is required for its regulation in response to luminal fructose loads (Patel et al., FASEB J 2015). We are currently looking at these potential mechanisms as candidate pathways.

We have modified the text in the Discussion to include the following in order to address the potential importance of the Txnip/GLUT interaction:

“The molecular mechanism through which Txnip regulates fructose absorption by the jejunum remains to be defined, and it is plausible that Txnip is necessary for the increase in fructose absorption in diabetic mice but is not directly causal. […] Regulation of GLUT2 and GLUT5-mediated fructose absorption could occur through a similar mechanism in which the binding of Txnip is necessary for this process.”

2) Can the authors comment on how the observed intestinal specificity might be conferred and what is the relative contribution of GLUT2 vs. GLUT5? (Reviewer 2)

As demonstrated in our findings, the deletion of Txnip only appears to affect fructosetransport through the intestine because there are no differences in the presence of 14C-labeled fructose in peripheral organs following an intravenous injection through the tail of mice. While we have no data to support why this specificity exists, this phenomenon supports the idea that Txnip may be acting as a scaffolding or recruiting protein in the membrane remodeling process through which the glucose transporters fuse with the brush border membrane. Because this remodeling process for both GLUT2 and GLUT5 is only known to occur in enterocytes in response to luminal sugar loads, the effect of Txnip in mediating this process may occur specifically in the small intestine.

A recent published study hypothesized that GLUT5 may be the primary fructose transporter in the gut as they observed an increase in GLUT5 mRNA expression in response to short-term fructose feeding while this difference was absent for GLUT2 (Patel et al., FASEB J 2015). Our results demonstrate that both GLUT2 and GLUT5 mRNA increase with long-term feeding, suggesting that under chronic conditions, both transporters contribute to the fructose transport process.

In a new experiment, we tested cellular 14C-labeled fructose uptake in Caco-2 cells overexpressing Txnip with GLUT2 and/or GLUT5. We found that overexpression of all three proteins produced the most significant increase in cellular fructose uptake in comparison to the empty vector transfected controls. When normalizing the uptake in cells overexpressing Txnip/GLUT2 or Txnip/GLUT5 to that of cells overexpressing all three proteins, we found that the normalized ratios of uptake were not different in cells overexpressing Txnip/GLUT2 and Txnip/GLUT5, suggesting that the effects of GLUT2 and GLUT5 may be similar and synergistic in Txnip-mediated fructose uptake in this cellular model system.

In order to include the quantification of the relative contribution of GLUT2 and GLUT5 in Txnip-mediated fructose transport, we have updated Figure 2 and the following text in the Results:

“Having established a molecular interaction between Txnip and both transporters that mediate fructose uptake, we then used Caco-2 cells transiently transfected with human Txnip, GLUT2, and GLUT5 overexpressing plasmid to determine if Txnip expression affects the cellular uptake of 14C-radiolabeled D-fructose. […] After normalizing the uptake in cells overexpressing Txnip/GLUT2 or Txnip/GLUT5 to that of cells overexpressing all three proteins, we found that the normalized ratios of uptake were not different in cells overexpressing Txnip/GLUT2 and Txnip/GLUT5, suggesting that the effects of GLUT2 and GLUT5 may be similar and synergistic in Txnip-mediated fructose uptake.”

*3) Please address the following issues: Figure 2 presents data showing decreased label (14C) in several tissues of Txnip KO mice after gavage of labeled fructose at a single time point. How was this time point determined? Given the rapid metabolic rate in mice, substantial compartmental flux likely takes place over this time period. The fructose would be metabolized and signals might not necessarily reflect fructose transport. The authors address this problem (and its relevance to the strep-induced state) with the data of Figure 5—figure supplement 2. But these data do not quantitatively account for fructose absorption, this is done at 10 minutes after oral gavage, and the results are confusing. The text in the subsection “Insulin-deficient diabetes induced by streptozotocin increases Txnip expression and fructose absorption” indicates that these findings represent stable isotope label from jejunum, but the legend of Figure 5—figure supplement 2 indicates that the signal is from jejunum and hepatic portal vein blood. Why pool these compartments?*

Thank you for this important comment. We determined the single time point for the data displayed in Figure 2 from a pilot gavage experiment conducted in C57BL6 wild type mice. We measured the 14C signal in peripheral blood following several time points and used 20 minutes as it was the time point with the maximum signal following the oral gavage, as illustrated in Figure 2—figure supplement 2 and in our revised text in the Results.

“To evaluate if the reduction in small intestinal fructose uptake caused by the deletion of Txnip regulates the levels of fructose in the peripheral bloodstream and other organs in vivo, we performed an oral gavage of 14C-radiolabeled D-fructose in WT and Txnip-KO mice. […] We used 20 minutes as the standard time point for this experiment, as it was the time point at which the peripheral blood in wild type mice contained the greatest 14C signal (Figure 2—figure supplement 2), which was also confirmed in our later experiments (Figure 5).”

Because Figure 2 is concerned with circulating levels of 14C-labeled fructose after transport from the small intestine, we found this time point suitable for this experiment. We confirmed this finding in the time course oral gavage experiment illustrated in Figure 5.

In order to address the fact that fructose may be quickly metabolized over this time course, we performed the stable isotope experiment illustrated in Figure 5—figure supplement 2. In this experiment, we used the 10 minute time point because this is the time point with the greatest significant difference between Txnip null and wild type mice for the jejunum. The figure legend contained a typo in which we indicated that the data represents measurements from both jejunum and hepatic portal vein blood, but the data, in fact, only represents stable isotope measurements from the jejunum. We apologize for the confusion and have corrected the figure legend.

While the 14C signal observed at later time points of the oral gavage experiment may represent a combination of both D-fructose and its downstream metabolites, fructose-specific metabolites, such as glyceraldehyde, can increase fructose absorption (Patel et al., FASEB J 2015). As such, we do not believe the rapid metabolism of fructose negatively affects our time point selection because D-fructose levels are significantly different in our experimental groups during the earlier time points of fructose absorption. In addition, the downstream metabolites of D-fructose may be partially responsible for the metabolic phenotype observed as a result of long-term fructose feeding because these may be more readily usable by other organs.

*4) Please address the following: Confocal images in Figure 4 purporting to show greater interaction between Txnip and the glucose transporters with the FSD could be more compelling. This interaction is confirmed using FLIM, but these changes are not robust, even if they are statistically significant. Is it possible that this interaction is due to a trivial reason, the presence of increased glucose transporter protein mass? Figure 4 presents data showing that FSD increases message for Glut2 and Glut5. Are Glut2 and Glut5 protein abundances increased? If so, more interaction would be expected. What is missing from this line of research is insight into how Txnip might promote fructose transport. At least for other transporters, protein mass is not the major predictor of facilitated movement of sugars across membranes. Instead, the fusion of transporter vesicles with the plasma membrane is critical. Does Txnip mediate this fusion event?*

We believe that the FLIM data represents a biologically significant change in the interaction between Txnip and GLUT2 and GLUT5, though the manner in which the data is presented and discussed was unclear, so we have clarified the text as indicated below.

We observed that the lifetime of interacting molecules of Txnip with the glucose transporters (τ_1_) decreases for both glucose transporters in fructose fed mice. As stated in the updated text, τ_1_ indirectly represents the distance between two molecules, since the donor signal, which in this case is Txnip, is quenched and has its lifetime decreased when the acceptor signal (GLUT2 or GLUT5) is in closer proximity. As such, the change observed may represent a difference in the nature of the interaction and can support the hypothesis that Txnip may be acting as a scaffold protein in the process of vesicle fusion of the glucose transporters with the membrane and thereby facilitating fructose transport.

τ_1_ decreases for both GLUT2 and GLUT5, indicating that fructose feeding changes the interaction between Txnip and both fructose transporters. As indicated by the Caco-2 fructose uptake experiment (Figure 2), both GLUT2 and GLUT5 contribute equally to fructose uptake. The interaction data acquired from the FLIM results support this result. The numerical changes of the τ_1_ values is actually quite strong, and the biological significance does not need to show a greater decrease. In fact, τ_1_ determines the relationship between the domains of the two molecules being probed by the antibodies, so, these changes may reflect a component of a more dramatic change in the relationship between Txnip and GLUT2 or GLUT5, or be sufficient by itself to impact the functional interaction in a major way.

If GLUT2 and GLUT5 protein abundances are increased, the FLIM measurements are not affected by the total protein mass visualized by immunofluorescence. Across all samples regardless of the changes in protein expression, we measured an equal number of FLIM values at individual co-localizing sites of Txnip and GLUT2 or GLUT5 molecules, so protein mass is unlikely to affect these events. We have modified the text to make this explicit.

Modification to text in response:

“We then analyzed the interaction between Txnip and GLUT2 or GLUT5 using Fluorescent Lifetime Imaging Microscopy (FLIM). Because two separate groups of mice were studied (normal diet and fructose diet) to determine the effect of fructose on the interaction between Txnip and GLUT2 and GLUT5, we analyzed FLIM values between Txnip and GLUT2 and GLUT5 within each group relative to donor only controls (Txnip only). […] Collectively, these data suggest that Txnip expression and interaction with GLUT2 and GLUT5 in the jejunum are responsive to chronic fructose consumption, and these changes may contribute to fructose absorption into the body.”